# Evaluation of Soil Infiltration Variability in Compacted and Uncompacted Soil Using Two Devices

Ján Jobbágy [1], Koloman Krištof [1,*], Michal Angelovič [1] and József Zsembeli [2]

1   Institute of Agricultural Engineering, Transport and Bioenergetics, Faculty of Engineering, Slovak University of Agriculture in Nitra, Trieda Adreja Hlinku 2, 949 76 Nitra, Slovakia; jan.jobbagy@uniag.sk (J.J.); michal.angelovic@uniag.sk (M.A.)
2   Karcag Research Institute, Hungarian University of Agriculture and Life Sciences, H-5300 Karcag, Hungary; zsembeli.jozsef@uni-mate.hu
*   Correspondence: koloman.kristof@uniag.sk; Tel.: +421-37-641-4368

**Abstract:** Infiltration is defined by the expression of the hydraulic conductivity of the soil, which we decided to monitor on an experimental field applying a modern system of land management (control traffic farming). The present study compared two different methods of monitoring the hydraulic conductivity of soil on a selected 16 ha plot of land in the suburbs of the village Kolíňany (Slovak Republic). During the growing seasons, crops such as oilseed rape, winter wheat, spring barley, winter barley, spring peas, and maize alternated in individual years. In addition to the study of hydraulic conductivity, a long-term experiment is underway to investigate the influence of technogenic factors on soil degradation using a system of controlled movement of machines in the field. A mini disk infiltrometer (method one) was used to evaluate the unsaturated hydraulic conductivity of the soil, and a double ring infiltrometer (method two) was used to measure the saturated hydraulic conductivity. Monitoring changes in soil infiltration capacity within the compacted and uncompacted lines required 26 monitoring points (20 for method one and 6 for method two). The first longitudinal line was compacted by an agricultural machinery chassis, and the second line remained uncompressed. The research also created transverse compacted lines at eight monitoring points (six for method one and two for method two). The results did not show a statistically significant difference when examining the effect of soil infiltration monitoring (compacted $p = 0.123$; uncompacted $p = 0.99$). When evaluating the statistical dependence on the compression caused by machinery in the track line, the hypothesis of significance was not confirmed ($p = 0.12$, at the level of 0.05). However, the results showed variability in the value of the difference factor between the two methods, ranging from 0 to 0.24. On average, it can be concluded that the results achieved using the double ring infiltrometer were 4.16 times higher than those measured with the mini disk infiltrometer. The variability of hydraulic conductivity was demonstrated not only in the compacted but also in the non-compacted part of the plot. In some places, the phenomenon of water repellency appeared, which could be caused by the drier location of the targeted plot.

**Keywords:** control traffic farming; water; sustainability; climate change; hydraulic conductivity

## 1. Introduction

The process of water entering the soil is called infiltration, most often through its surface. From a hydrological point of view, the most interesting is infiltration from precipitation. The formation of surface runoff and the associated soil erosion depends on the intensity of the infiltration. Experts' goal is to create such conditions that as much rainfall as possible needed for plants soaks into the soil [1]. As already mentioned, infiltration is the process of water infiltration into the ground. In addition to the classic cases of water infiltration from precipitation, irrigation or melting snow, water can infiltrate the soil from various standing waters such as puddles, lakes, and reservoirs. Other possibilities are through banks, the bottom of streams, or infiltration facilities below the soil surface [2].

One of the crucial information points for effective and comprehensive water management is the rate of soil infiltration, which determines the proportion of infiltrated water and (drainage) runoff. Once the soil is fully saturated, the infiltration rate becomes constant (i.e., a steady state) and equals the hydraulic conductivity of the saturated soil. Along with evapotranspiration, infiltration contributes to continuous precipitation losses. It is one of the decisive factors determining the water balance of urban river basins, especially in the case of large open spaces [3]. The infiltration rate in terms of pedology is how soil can absorb water from rainfall or irrigation. It is measured as the displacement rate in mm of a water column per time unit or millilitres per unit time [4]. The infiltration rate is expressed as the ratio of the amount of water infiltrated over a unit of soil surface area per time unit [2]. Hydraulic conductivity is used to model and dimension drainage systems and deploys rainwater control measures [5]. When using infiltration models, it is necessary to define different soil infiltration parameters, which is crucial in meteoric water and (drainage) runoff modelling. It significantly affects the distribution of precipitation into losses and effective rainfall [6,7]. Hydraulic conductivity has the most significant statistical variability between different soil hydrological properties and as it was stated [8–10], many factors can influence it (e.g., soil type, land use, landscape location, depth, tools, and assessment methods). This parameter is not constant and varies in time and space. In addition, implemented measures (e.g., plants) can create variability [11]. Soil heterogeneity can be induced even by small changes in soil physical properties (for low permeability clay species), such as cracks and canals [12].

Several instruments are used to measure and evaluate infiltration, which can be divided into two large groups. In the first, water is applied to the soil at positive pressure (e.g., a dual-circuit infiltrometer). In the second, water is applied at a negative pressure (e.g., a voltage infiltrometer) [8]. The two-circuit infiltrometer is a well-known and reliable field method for measuring the hydraulic conductivity of saturated soil [13]. However, it is time-consuming, unsuitable for sloping terrain. It requires a relatively large amount of water to perform the test, which is a limitation for areas with limited access to water [14]. The second group includes the modern method of measuring by the mini disk infiltrometer, which measures unsaturated hydraulic conductivity (practical field method, fast and low water demand of about 135 mL for one measurement) [15]. Water is infiltrated into the soil through a semi-permeable stainless-steel membrane, and the amount of water infiltrated into the soil is read on the scale. It is a smaller version of the voltage infiltrometer, which uses voltage (low vacuum) to prevent the applied water from entering the macropores. However, various sources suggest that the results could vary significantly [16].

Other sources provide three basic methods of measuring infiltration, the tank method (water usually infiltrates in two concentric cylinders or large infiltrometers of different floor shapes), the strain gauge method (in this infiltration measurement method, water passes through a low permeability porous plate), and the method of irrigation (infiltration measurements are performed indirectly) [17]. Another method is a block furrow infiltrometer, or more precisely, instruments that measure inflow and outflow [18].

Based on the summarised overview and the available technique at the workplace, it was decided to characterise the spatial variability of the infiltration capacity of the soil in a compacted and non-compacted soil track using two devices on a selected agricultural plot. The aim of the paper was therefore to monitor the variability of unsaturated and saturated hydraulic conductivity during the season on the selected plot, while the results were subjected to statistical analysis. From the obtained results, the following hypotheses were assumed:

- The method used to determine infiltration in compacted and non-compacted soil track is statistically significant.
- The selected research area in terms of the degree of compaction has a significant effect on the degree of infiltration capacity of the soil.
- The phenomenon of "water repellency" can be alleviated by extending the measurement time.

As part of the research carried out by the mini disk infiltrometer for the analytical determination of similarities or differences between individual sampling sites (compacted and non-compacted), the results were subjected to further statistical analysis, namely the method of hierarchical clustering. The output of the mentioned method is a dendrogram, which determines the degree of difference in the obtained results.

## 2. Materials and Methods

### 2.1. Locality Characteristics

The targeted land with an area of 20 ha is located in the rural area of agricultural land in Kolíňany (Jelenecká, 48°22′06.28″ N, 18°12′26.67″ E, altitude from 191 to 209 m a.s.l., Figure 1).

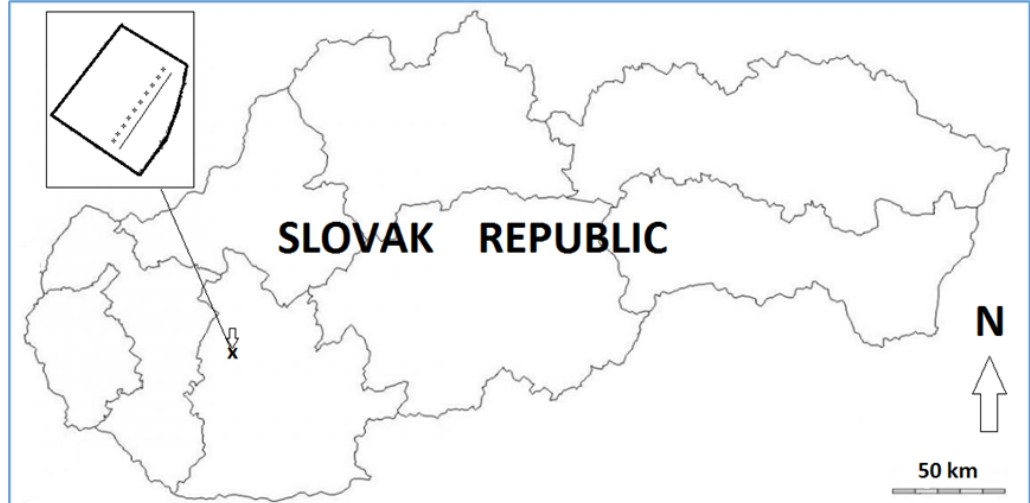

**Figure 1.** Location of the measurements (X—exact location of experimental plot; detail—the shape of experimental plot and layout of measurements across the plot).

The area was monitored for a longer period (more than ten years) during which research work was carried out on it (input information—an experiment based on spring barley sowing in 2009, the plot was deeply ploughed in autumn 2008), during which crop cultivation was introduced with the deployment of the controlled movement of machines across the field (Figure 2). During various experiments, crops were grown on the plot, according to the management of the VPP Kolíňany farm enterprise. Agrotechnical terms, fertilisation, and stand treatment (including applied doses) were solved based on the decision of the agronomist of the company. Thus, according to BPEJ (Slovak national standard for identification of soil production potential) 0150002, the land is classified to be located in a warm, dry, and lowland climate region. The soil structure class at the top (0–350 mm) is heavily loamy (51% clay, 30% sand, and 19% silt). These are medium–heavy soils (loamy soil) and brown pseudogley soil. Records show that it is a plain without the manifestation of surface water erosion (Table 1). Technological operations were performed according to the current needs of the cultivated crop and growing conditions. Tillage took place without ploughing, using plate and coulter tools up to a depth of 15 cm. All auxiliary operations (filling and emptying of tanks) were performed at the dead centre. Plant residues were crushed and incorporated into the soil. The controlled movement of machines with a working width module of 6 m (and its multiples) was purposefully observed on the plot (Table 2). In the experiment, the area without machine passage (non-compacted soil 64%) and the area with machine passage (pressed soil 34%) were provided. All targeting and motion control are solved using GNSS (GPS) technology. For the possibility of an exact assessment of the impact of soil compaction, three transverse belts with a width of 25 m were created on the plot by passing the tractor (wheel next to the wheel) in the period immediately after harvest, i.e., only once a year. The results of research on crop variability, soil strength, and other soil properties have been reported in various publications [19,20].

The average total precipitation is (1989–2018) 579.1 mm, and the average temperature reaches 10 °C, ranging from −12 °C in winter to 38 °C in summer [21,22].

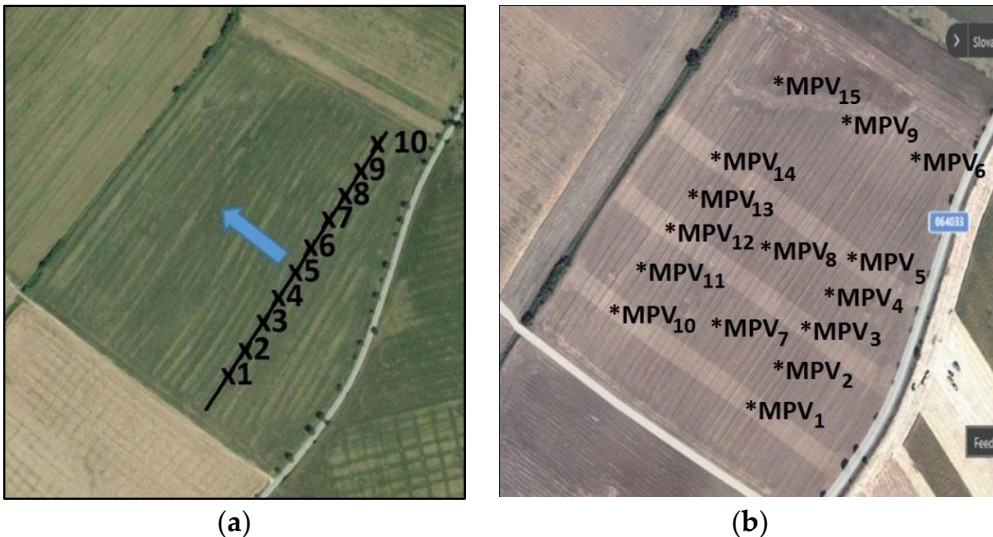

| (a) | (b) |
|-----|-----|

**Figure 2.** Experimental field in Kolíňany (figure on the right, reprinted/adapted with permission from Ref. [19]. 2014, ProfiPress ©. [19]), (**a**) Monitoring points for determining the spatial variability of infiltration on the plot, numbers reflects the measurements order and arrow is the measurement replication direction (**b**) Monitoring points to determine the variability of infiltration in the compacted and uncompacted line (where * is the exact measurement point location, $MPV_{1-15}$ are code identifiers, where brown colour difference represents compacted and uncompacted plot zones).

**Table 1.** Soil and weather properties.

| Soil Properties | | |
|---|---|---|
| **Code** | **Code Description** | **Properties** |
| 01 | Climate region code | warm, dry, lowland |
| 50 | Main soil unit code | pseudogley brown soil on loamy and polygenic clays, medium–heavy |
| 0 | Slope and exposure code | plain without surface water erosion |
| 0 | Soil skeleton and depth code | deep soil without skeleton |
| 2 | Soil grain size code | medium–heavy soils (clay) |
| Soil texture | | |
| Clay | | 51% |
| Sand | | 30% |
| Silt | | 19% |
| Weather Properties | | |
| Precipitation, mm | Average (1991–2020) 521 | Sum, 2020 578.4 |
| Temperature, °C | Average (1991–2020) −2 ÷ 24 | Range, 2020 −4 ÷ 27 |

Note: Adopted and summarised from publicly available database [23,24].

Soil texture indicates the content of the three main size categories of "fine earth" in the soil according to their weight proportions, which are sand, silt, and clay, with specific particle sizes for sand (0.05–2 mm), silt (0.002–0.05 mm), and clay (<0.002 mm) [25].

**Table 2.** Experimental field characteristic—average yields and soil cone index (Reprinted/adapted with permission from Ref. [26]. 2022, © by the authors. Licensee MDPI, Basel, Switzerland. [26]).

| Crop Variability | | |
|---|---|---|
| **Harvest Years** | **Crop** | **Average Yield (t·ha$^{-1}$)** |
| 2009 | Spring barley cv. Kango | 5.0 |
| 2011 | Winter wheat cv. Augustus | 6.2 |
| 2014 | Spring barley cv. Kango | 4.8 |
| 2016 | Winter wheat cv. HYFI | 7.9 |
| 2017 | Winter barley cv. Wintmalt | 6.7 |
| 2019 | Winter wheat cv. RGT Reform | 7.8 |
| 2021 | Spring barley cv. IS Maltigo | 3.0 |
| Soil cone index (soil strength) | | |
| **Zone \*** | **Variability, MPa** | |
| A | 0.59–2.5 | |
| B | 0.78–2.35 | |
| C | 0.9–3.45 | |

Notes: * A—no traffic; B—single pass; C—multiple passes/permanent tramlines [26].

### 2.2. Characteristics of Measuring Technology

Our research work was primarily aimed at evaluating the variability of soil infiltration capacity. In the first case, we focused on the evaluation of spatial surface variability, where we chose 15 monitoring points as a starting point. The measurement of infiltration and its evaluation was carried out in two time cycles of 300 s and 900 s. Monitoring points in the first case were concentrated according to previous research, in the parts compressed in the transverse direction (MPV$_1$, MPV$_3$, MPV$_5$, MPV$_{10}$, MPV$_{12}$, MPV$_{14}$) and outside them. ArcGis (ArcView 3.2, ESRI, Redlands, CA 92373-8100, USA) and Spline interpolation were used to evaluate the results. In addition, we decided to monitor infiltration also in the longitudinal direction through saturated and unsaturated hydraulic conductivity.

Our research focused on evaluating the variability of soil infiltration capacity through saturated and unsaturated hydraulic conductivity. The total land area was 16 ha, and the crop grown in the year of research was wheat. Ten fixed points were selected on the plot (Figure 2, positions x$_1$ to x$_{10}$), where ten monitoring points were spaced from the slope, perpendicular to the auxiliary line (track) at a distance of 0.7 m (compacted part, wheels of various agricultural machinery approx. ten times a year) and another ten points were at a distance of 2.7 m (uncompacted part) (i.e., a total of twenty monitoring points for the compacted and uncompacted lines—applies to the first measurement method one). Points x$_2$, x$_4$, and x$_6$ were selected in long-term compression zones of the tractor wheels (every year, a John Deere 8230 tractor is used, JOHN DEERE, Moline, IL 61265, USA, wheel width 0.62 m and with an inflation pressure of 0.2 MPa) by the track-by-track method (Figure 2 right). For examining hydraulic conductivity with method two (double ring), six monitoring points were selected at fixed points x$_1$ to x$_3$ (three points at a distance of 0.7 m—compacted part, and three at a distance of 2.7 m—uncompacted part). The distance of 0.7 m represents a line for driving machinery (just before the measurement with the Claydon Hybrid T6 seed drill – CLAYDON, Wickhambrook, Newmarket, Suffolk, CB8 8XY, UK); the distance of 2.7 m represents a route that was not compacted at all. Thus, unsaturated hydraulic conductivity measurements were performed at 20 monitoring points with a mini disk infiltrometer (Decagon Devices, Pullman WA 99163, USA, Figure 2a). Due to higher water consumption and time, saturated hydraulic conductivity measurements were only performed at six points using a double ring infiltrometer (Double Ring, Eijkelkamp, 6987 EN Giesbeek, The Netherlands) (Figure 2, x$_1$, x$_2$, and x$_3$).

The soil structure determines the adaptation parameters for data analysis using the two proposed measurement methods. In addition, knowledge of soil structure can provide an approximate view of the expected hydrological properties. It can also determine other

infiltration model parameters and a control and reference for data analysis, results, and interpretation [27].

The Topcon GRS-1 handheld satellite navigation device was used to determine the boundaries of the plot (RTK accuracy, corrected signal: SKPOS cm, Figure 2b, Topcon, Livermore, CA 94550, USA, Figure 3) [28]. The measured section was oriented approx. 40 m from the road along the contour of the soil horizon. The values were recorded every 30 s for measurement method one and every 60 s for method two (30 values were recorded). A more detailed description of the equipment and methodology for the determination of unsaturated and saturated hydraulic conductivity is given in the following text.

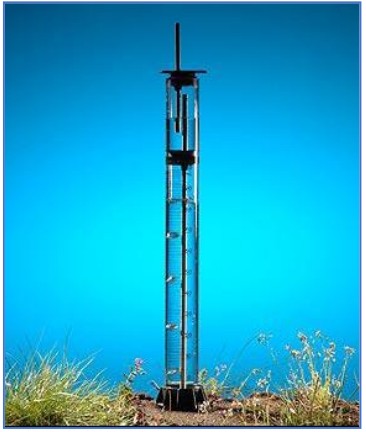 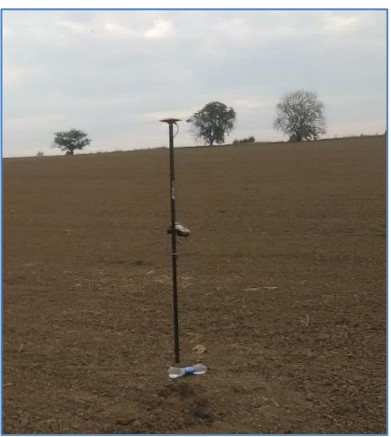 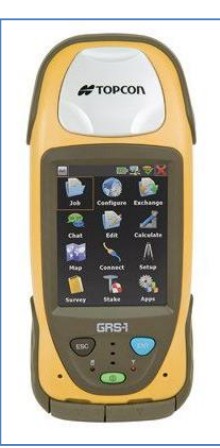

**Figure 3.** Mini disk infiltrometer and GPS Topcon device.

*Mini disk infiltrometer* (Figure 3) *method*. In the case of infiltration monitoring, this is an easier and faster way to assess the infiltration capacity of the soil. It is a practical version of a voltage infiltrometer for monitoring unsaturated hydraulic conductivity ($k$), especially in cases where the access to water and in sloping terrain is limited. The mini disk infiltrometer consists of a polycarbonate tube with a diameter of 31 mm and a height of 327 mm (with a volume of 135 mL) [29]. The tube is divided into two chambers filled with water. The upper (bubble) chamber adjusts the air intake using an adjustable suction control steel tube (0.5–7 cm, depending on the soil type). Water from the bottom water reservoir chamber infiltrates into the soil through a semi-permeable stainless-steel membrane (porous sintered stainless-steel disk) located at the bottom of the tube. The lower part is equipped with a scale in millilitres, from which the volume of infiltrated water is read. We performed measurements according to the methodically given procedure. A program for graphical processing of results is also available for the infiltrometer.

Soil hydraulic conductivity is defined as "how many meters of water per day seep down into the soil, by gravity or by a unit of pressure drop" [30], which can be determined with various methods. One of them calculates soil infiltration and hydraulic conductivity determined by Zhang [31]. This method requires the measurement of cumulative infiltration over time, and the determination of the result is by function (Equation (1)). The results of the measurements are plotted, where the $x$-axis represents the square root of time and the $y$-axis the cumulative infiltration. Subsequently, the data points are equipped with a function (e.g., the first two terms of the infiltration equation).

$$I = C_1 t + C_2 t^{1/2} \tag{1}$$

where $I$ (m) is the cumulative infiltration, $C_1$ (m·s$^{-1}$) is a function parameter related to hydraulic conductivity, $C_2$ (m·s$^{-1/2}$) is a function parameter related to the sorption capacity of the soil. The soil hydraulic conductivity $k$ is calculated using the following function:

$$k = \frac{C_1}{A} \tag{2}$$

where $C_1$ is the slope of the cumulative infiltration curve as a function of the square root of time, $A$ is the van Genuchten parameter, which is based on the air intake tube setting (suction rate) for the soil type and the radius of the infiltrometer disk, specified by the manufacturer. It is also calculated according to the following equation:

$$A = \frac{11.65(n^{0.1} - 1)exp[b(n - 1.9)\alpha h_0]}{(\alpha r_0)^{0.91}} \quad (b = 2.92 \; if \; n \geq 1.9; b = 7.5 \; if \; n < 1.9) \quad (3)$$

where $n$ and $\alpha$ are van Genuchten parameters determined for 1 of the 12 soil texture classes (from sand to clay), $r_0$ is the disk radius, $h_0$ is the suction at the disk surface, and $b$ is a constant. The mini disk infiltrometer infiltrates water at the suction of $-0.5$ to $-6$ cm and has a radius of 2.2 cm. The van Genuchten parameters for the 12 texture classes were obtained from Carsel and Parrish [32].

According to the latest methodologies for measurements on soils with a value of $n$ less than 1.35, it is recommended to change the original equation [33] to a modified equation [30], thus improving the estimates of $k$. The Kutilek and Nielsen method [34] was used to estimate $k_s$ (cm·s$^{-1}$) from mini disk infiltrometer measurements based on Equation (1):

$$I \approx mk_s t + C_2 t^{1/2} \quad (4)$$

where $C_2$ (m·s$^{-1/2}$) is a function parameter related to the sorption capacity of the soil and $m = 0.667$.

*Double ring infiltrometer* (Figure 4) *method*. The standard double ring infiltrometer set consists of three inner and outer rings, a driving plate, an impact-absorbing hammer, measuring bridges, and measuring rods with floats. The pairs of stainless-steel infiltration rings have the following diameters: 28/53 cm, 30/55 cm, and 32/57 cm (we applied the two first sets). The ring's height is 25 cm, and it has one cutting edge. The purpose of the outer ring is to have the infiltrating water act as a buffer zone against infiltrating water straining away sideways from the inner ring (this applies in particular to heterogeneous soils) [35].

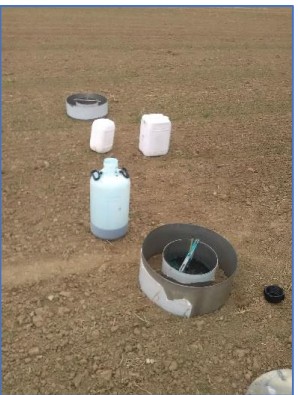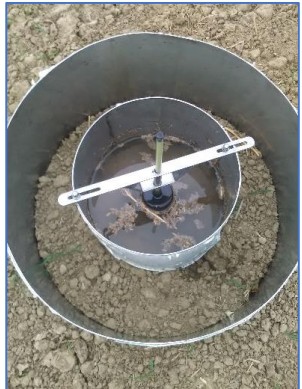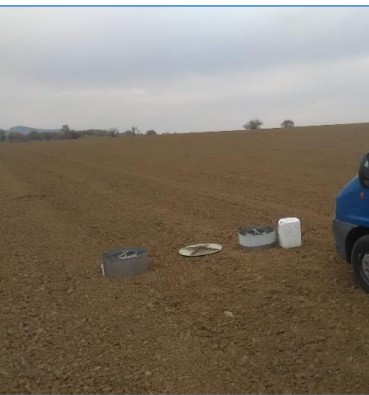

**Figure 4.** Double ring infiltrometer measurement.

Pull-out hooks are also available for the double ring infiltrometer set. The inner rings are fitted with a synthetic measuring bridge with a measuring rod and float. The rod easily moves up and down through a small tube in the measuring bridge. The float is oriented in the middle of the rings and, together with the rod, indicates the water level. Infiltration measurements are used in the field, where this device is used to determine the saturated hydraulic conductivity of the soil $k_s$. The method is based on inserting the inner ring into the outer ring of the prescribed diameter. The placement depth also depends on the compacted soil surface layer. After fitting the rings, the principle is flooding them with water and then monitoring the drop in the water level in the inner circle over time. The process is repeated until the soil is saturated, i.e., the quasi-steady infiltration rate is the same for two consecutive tests. The degree of infiltration results from the inner diameter

of the circle, a certain drop in level, and the time recorded for each step. Filling one set requires up to 25 L of water (we applied 10 L of blue rock concentrate), and the circles should be filled to 5 to 10 cm. To obtain optimal infiltration results, it is advisable to use water of similar quality and temperature as in the measured environment. The sequence of the depreciation of values for the determination of saturated hydraulic conductivity and infiltration rate was 60 s. Vertical infiltration was calculated according to the relationship (based on Philip's known infiltration equations) [35]:

$$i = St^{1/2} + A_1 t \tag{5}$$

where $i$ (cm) is cumulative infiltration, $t$ (s) is time, $A_1$ (cm·s$^{-1}$) is steady-state infiltration rate, and $S$ (cm·s$^{-1/2}$) is sorptivity.

The infiltration rate $v$, therefore, is [36]:

$$v = \frac{1}{2}St^{-1/2} + A_1 \tag{6}$$

To evaluate the measured data, it was necessary to determine the parameters $S$ and $A$. One of the most commonly used approximation methods is the "least squares" method. We use the found parameters to calculate the infiltration rate at steady-state and saturated hydraulic conductivity $k_s$. The equilibrium state of infiltration after a long time interval of infiltration remains constant, and its values approach the values of saturated hydraulic conductivity [37,38]:

$$k_s = \frac{A_1}{m} \tag{7}$$

where $k_s$ (m·s$^{-1}$) is hydraulic conductivity and $m$ is constant, 0.6667.

Chromíková et al. used the same relationships to determine the infiltration rate and saturated hydraulic conductivity [39].

### 2.3. Statistical Evaluation of Results

The applied methodologies for determining the state of infiltration in the compacted and uncompacted soil track assume that the results will not be without statistical significance. The obtained results are followed by demonstrable or unprovable changes in monitoring the impact of the compacted and uncompacted soil and, on the other hand, monitoring the research methodology used. Statistical analyses must be used for quality and adequate evaluation of results.

A one-factor ANOVA test was used to test the significance of the differences between the measurements performed on the compacted and uncompacted soil and in the following to compare the two different periods. A one-factor ANOVA test (Statistica ver. 2009, TIBCO Software, Palo Alto, CA 94304, USA) was used to test the significance of differences between measurements performed on compacted and uncompacted soil (to evaluate and compare the results of compacted and uncompacted soil (Formula (8)) but also to evaluate the impact of the measurement method (Formula (9)):

$$y_{ij} = \mu + S_i + e_{ij;\ \mathrm{mm}} \tag{8}$$

$$y_{ij} = \mu + M_i + e_{ij;\ \mathrm{mm}} \tag{9}$$

$y_{ij}$—measurement parameter;
$\mu$—overall mean;
$S_i$—effect of the soil;
$M_i$—effect of the methods;
$e_{ij}$—Random error with mean 0 and variance $\sigma^2$.

The hierarchical clustering method (GitHub, Inc., Orange software; Orange, Paris, France, 2023; available free: https://orangedatamining.com/download/#windows, ac-

cessed on 26 March 2023) was used for the analytical determination of similarities or differences between individual sampling points (compacted and uncompacted, mini disk infiltrometer sampling). The output is the so-called dendrogram for any object based on a distance matrix. Distances are calculated using Euclidean metrics, the ordinary direct distance between two points in a Euclidean space.

Thus, distances represent differences in measured values between rows where the method of map distances has been applied (it visualises distances between objects) by replacing the numbers given in the distance matrix with colour differences.

## 3. Results and Discussion

### 3.1. Spatial Variability of Hydraulic Conductivity

As part of the research activity of the surface variability of soil infiltration capacity monitored by hydraulic conductivity, it was found that the hydraulic conductivity reached a maximum value of more than $6 \times 10^{-4}$ cm·s$^{-1}$ at a 900 s time cycle. The time horizon of results monitoring affects the overall results achieved. At a 300 s cycle on an area of 9.28 ha (58%), the hydraulic conductivity reached values in the range of 1.53 to 2.72 cm·s$^{-1}$. In the case of the 900 s cycle, the area increased to 10.1 ha, with a margin slightly different from the first case. When focusing on the spatial variability within the compacted three bands (Figure 5), however, it can be observed that not all points demonstrated the significance of the reduction of hydraulic conductivity (e.g., MPV$_3$, MPV$_5$, MPV$_{10}$). In the case of monitoring the other surface, differences were also monitored, where the hydraulic conductivity reached a maximum of 0.85 cm·s$^{-1}$ (MPV$_2$, MPV$_4$). Higher values of hydraulic conductivity (above 2.54 cm·s$^{-1}$) were observed on an area of 0.7 ha. The values obtained at monitoring point 15 were interesting. In certain periods of time, excessive moisture was maintained on the mentioned area, even a wet part, while this water caused the formation of an excessively wet soil structure, which can also be seen from Figure 2. The average value of hydraulic conductivity was 1.74 cm·s$^{-1}$ with a coefficient of variation of 80.01%. The statistical analysis of the dependence on the time period (300 s or 900 s) at all 15 monitoring points shows significance ($p = 0.11$). Negative values were not observed during the research activity.

Spatial variability of infiltration was also dealt with by other authors. They pointed to the fact that even in a smaller area, the variability of the hydraulic conductivity of the soil is demonstrated. The linear interpolation method was also used by Radinja [40]. The results sometimes showed up to three-fold differences in the investigated variability. In another researched work, the authors focused on the surface variability of infiltration when applying different work operations, while in places the values even reached negative values (water repellency). In addition, the measurements made immediately before and just after the application of supplementary irrigation confirmed that the hydraulic conductivity of the soil was reduced by up to 51%. However, the latter measurements did not confirm the expectations of the authors, because there was no increase in the hydraulic conductivity, which should have been caused by the change in the soil structure by performing the subduction [41].

### 3.2. Results of Mini Disk Infiltrometer Measurements (Method One)

The measurements in the presented paper focused on evaluating the soil infiltration capacity expressed by unsaturated and saturated hydraulic conductivity. Table 3 shows the results achieved by the mini disk infiltrometer measuring device, while the hydraulic conductivity $k$ (cm·s$^{-1}$), standard deviation (cm·s$^{-1}$), and coefficient of variation $CV$ (%) were calculated. A total of 20 measurements were taken, the first 10 being in the compacted zone (0.7 m from the fixed points) and the remaining 10 in the non-compacted zone (2.7 m from the fixed points). The value of parameter $A$ was 2.8. Negative values were excluded in the calculation of the statistical results (highlighted in grey in Table 3) because they affected the value of the coefficient of variation (values exceeded 100%). Negative values arise due to the alternation of the change in infiltration between individual time intervals

to higher and lower values and vice-versa. It can be caused by the soil condition in the measured locality or incorrect measurements. The emergence of the phenomenon of water repellence caused by the dry area in the soil reducing the value of *k* can also be an explanation. This phenomenon has also been observed in other publications [42–44], which state that low water content and high temperatures cause soil water repellence. The negative effect can be mitigated, as it can be seen from our results with a longer measurement of infiltration (extension from 300 s to 900 s), representing an increased amount of water at the measurement site. The phenomenon was confirmed at all measured points, where negative values were found changing to positive values, or more precisely, by close-to-zero values rising closer to positive values. The results of other authors confirmed this [45,46], where they agreed that it was not a stable soil condition and could be mitigated by the inflow of water which increases the water content of the soil.

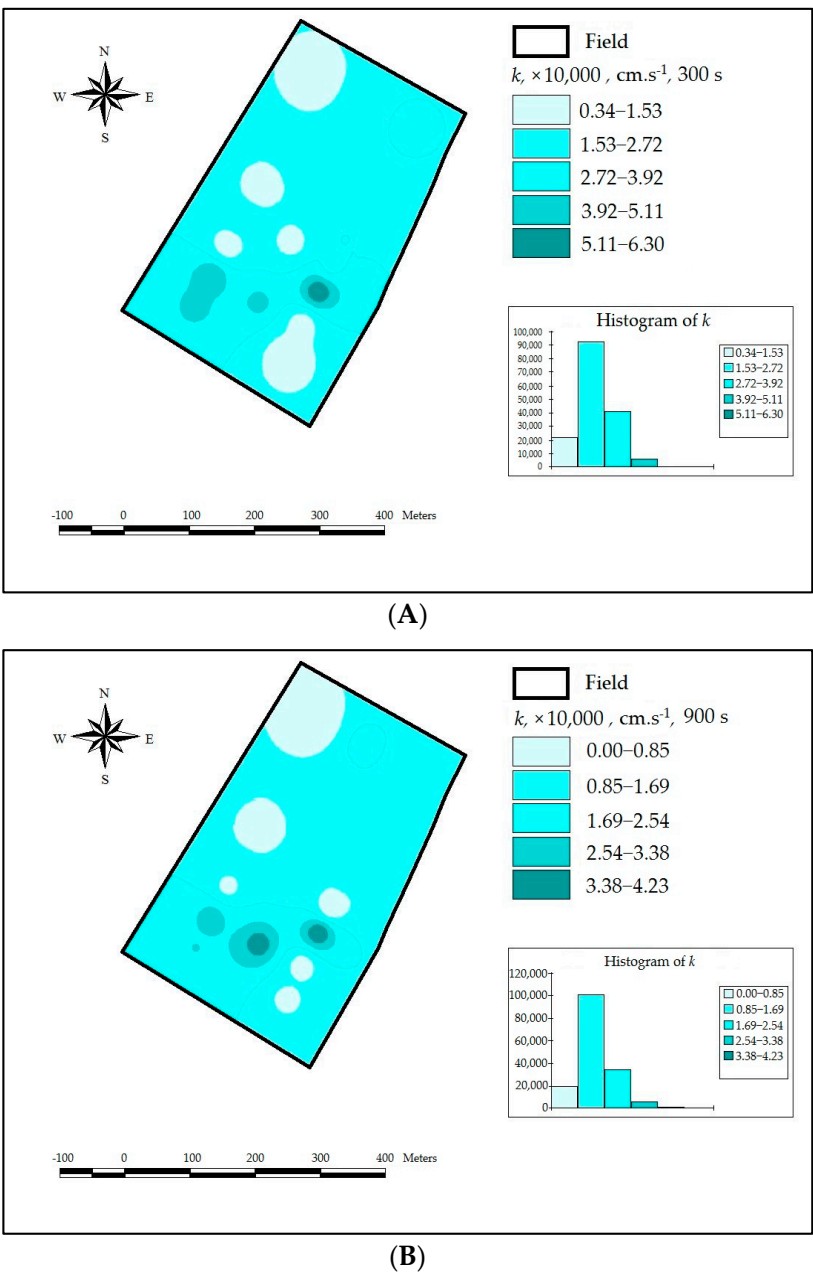

**Figure 5.** Spatial variability of hydraulic conductivity, (**A**) measurements of 300 s cycle, (**B**) measurements of 900 s cycle.

**Table 3.** Hydraulic conductivity measured with the mini disk infiltrometer in monitoring points at 0.7 m and 2.7 m distances.

| MP | $C_1$ | $k$, $\times10^{-4}$ cm·s$^{-1}$ | $C_1$ | $k$, $\times10^{-4}$ cm·s$^{-1}$ | $C_1$ | $k$, $\times10^{-4}$ cm·s$^{-1}$ | $C_1$ | $k$, $\times10^{-4}$ cm·s$^{-1}$ |
|---|---|---|---|---|---|---|---|---|
| | Measurement Data for MP—0.7 m | | | | Measurement Data for MP—2.7 m | | | |
| | T = 300 s | | T = 900 s | | T = 300 s | | T = 900 s | |
| $x_1$ | 0.0000 | 0.00 | 0.0003 | 1.07 | 0.0006 | 2.14 | 0.0005 | 1.79 |
| $x_2$ | −0.0003 | −1.07 | 0.0005 | 1.79 | 0.0000 | 0.00 | 0.0000 | 0.00 |
| $x_3$ | 0.0004 | 1.43 | 0.0000 | 0.00 | −0.0005 | −1.79 | 0.0001 | 0.36 |
| $x_4$ | 0.0000 | 0.00 | 0.0004 | 1.43 | −0.0006 | −2.14 | −0.0003 | −1.07 |
| $x_5$ | 0.0007 | 2.50 | 0.0003 | 1.07 | 0.0000 | 0.00 | 0.0004 | 1.43 |
| $x_6$ | 0.0004 | 1.43 | 0.0004 | 1.43 | −0.0008 | −2.86 | −0.0002 | −0.71 |
| $x_7$ | 0.0003 | 1.07 | 0.0005 | 1.79 | 0.0005 | 1.79 | 0.0002 | 0.71 |
| $x_8$ | 0.0001 | 0.36 | 0.0004 | 1.43 | 0.0011 | 3.93 | 0.0006 | 2.14 |
| $x_9$ | 0.0001 | 0.36 | 0.0003 | 1.07 | 0.0006 | 2.14 | 0.0000 | 0.00 |
| $x_{10}$ | 0.0001 | 0.36 | 0.0003 | 1.07 | 0.0006 | 2.14 | 0.0004 | 1.43 |
| Aver | | 0.83 | | 1.22 | | 1.73 | | 0.98 |
| St. dev. | | 0.79 | | 0.49 | | 1.27 | | 0.77 |
| CV (%) | | 94.59 | | 39.99 | | 73.37 | | 78.71 |

Notes: Statistical parameters excluding the negative values (indicated with grey colour), MP monitoring points ($x_1$–$x_{10}$), *CV*—coefficient of variation.

The results clearly show line variability within the monitoring points in the soil compacted by the seed drill and in the soil not compressed. The coefficient of variation ranges from 39.99 to 94.59%. Other authors have also reported significant spatial variability [40,41,47,48].

The information obtained shows that different soil types have different water infiltration rates. For some soil types, this can be problematic due to either high speed (sandy soils) or low speed (compact soils) [31].

Figure 6 shows the dependence of cumulative infiltration on the square root of time in the time interval of measurement 300 s ($x_1$, 0.7 m—compacted soil with a seed drill, 2.7 m—uncompacted). The first case is an almost continuous change of cumulative infiltration with the slope parameter of curve $C_1$ determined according to the methodology. After determining these results, the hydraulic conductivity (*k*) was determined. However, after increasing the number of measurements to 30 (900 s), the curve changed slightly (Figure 6, $x_1$, the value of the coefficient reached 0.0003, the parameter $R^2$ changed from 0.989 to 0.995).

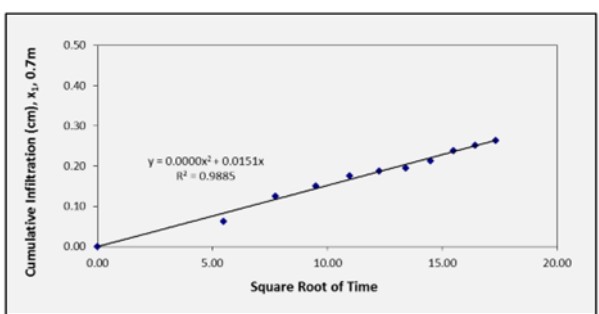 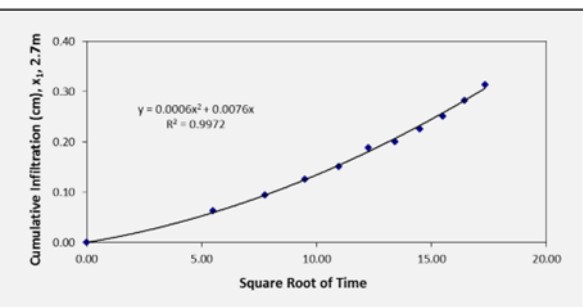

**Figure 6.** Dependence of cumulative infiltration on the square root of time ($x_1$, 0.7 m and 2.7 m, 300 s) determined with method one.

For the other monitoring points, the results were graphically evaluated only for the time interval of 900 s (Figure 7). The trend at the monitoring points (time interval 300 s, compacted line) was ascending up to two points, where the coefficients of the curve showed a stable, unchanging trend of continuous infiltration ($x_1$, $x_4$, $C_1$ = 0.0000); the trend up to one point was based on the measured data, a situation where a negative value occurred ($x_2$, $C_1$ = −0.0003). At longer study time horizons (900 s for each MP), the hydraulic conductivity was greater than 0 at almost all points (except $x_3$, where $C_1$ = 0.0000). The monitoring of unsaturated hydraulic conductivity shows that it is necessary to carry out

measurements for this type of soil and farming (management) at longer intervals. The coefficient of variation (*CV*) reached 39.99% in compacted soil in the range of hydraulic conductivity from 0 to $1.79 \times 10^{-4}$ cm·s$^{-1}$ (average value $1.22 \times 10^{-4}$ cm·s$^{-1}$). Compared to the shorter measurement time, where the *CV* value reached 94.59% (too high variability), the variability was up to 2.37 times higher.

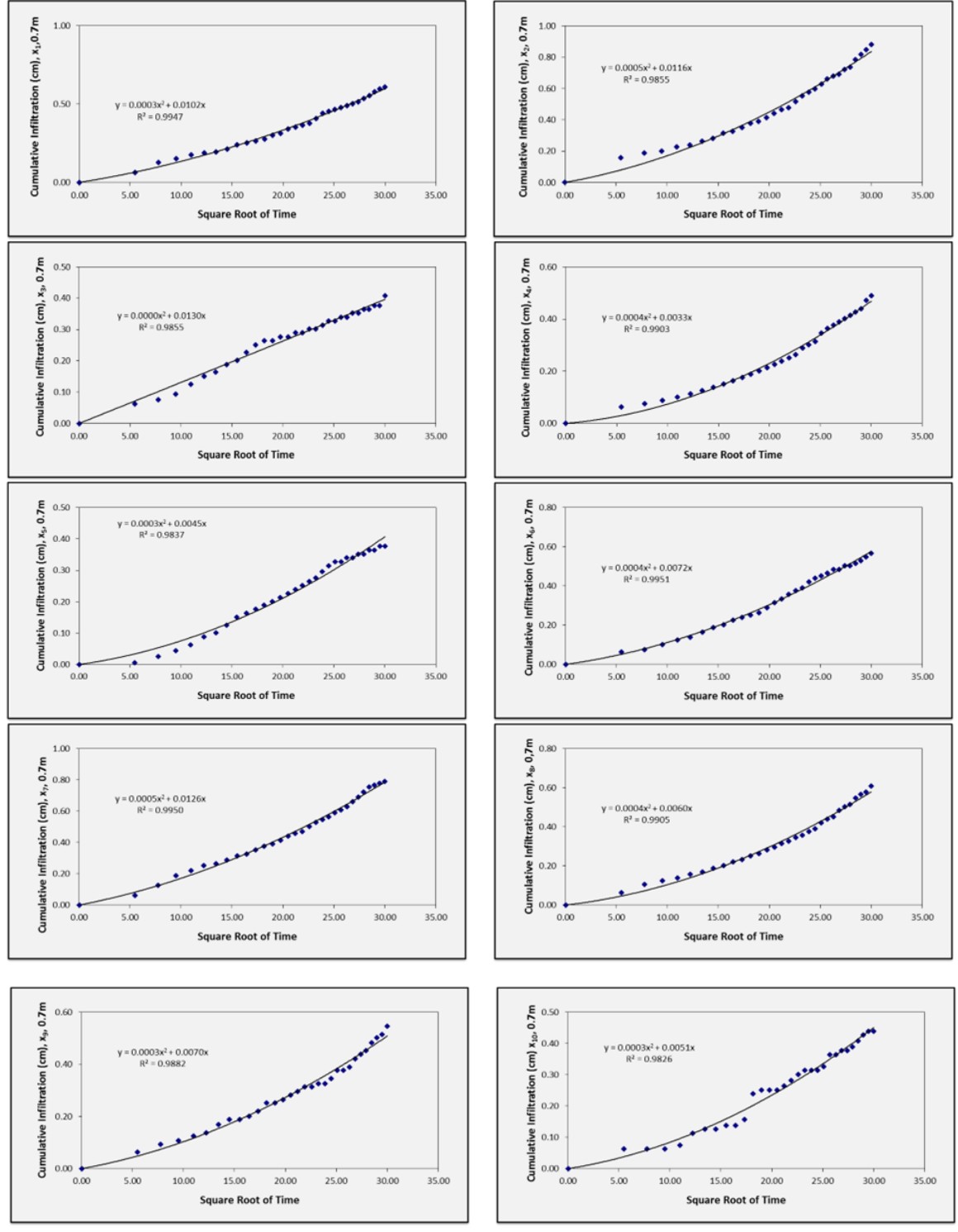

**Figure 7.** Dependence of cumulative infiltration on the square root of time (from $x_1$ to $x_{10}$, 0.7 m, measured time interval 900 s) determined with method one.

Figure 7 presents the results of cumulative infiltration as a function of the square root of time in the uncompacted locality of interest ($-2.7$ m from the auxiliary line, Figure 2). Hydraulic conductivity reached a variability of 78.71% with an average value of 0.98 cm·s$^{-1}$ (measurement time interval was 900 s). For the uncompacted line, the variability was from $-1.07 \times 10^{-4}$ to $2.14 \times 10^{-4}$ cm·s$^{-1}$, where we had two cases with negative values ($x_4$ and $x_6$, $C_1 < 0.000$) and two cases with a zero value ($x_2$ and $x_9$, $C_1 = 0.0000$). The graphical representation of the results (Figure 7) clearly demonstrates that the infiltration of water into the soil expressed by hydraulic conductivity is a complex phenomenon, and at some monitoring points, there were jumps and not smooth changes ($x_3$, $x_4$, $x_6$, and $x_9$).

Other authors also dealt with monitoring unsaturated hydraulic conductivity, whose results are presented through, e.g., the impact of tillage technology (no tillage, conventional tillage, and minimal tillage). The results showed deviations in conventional tillage with ploughing up to 30 cm compared to other technologies, where $k$ was 2.5 times lower [49]. One of the main reasons for integrating soil protection technologies into the management (farming) system is the decrease in melting the surface layer of the soil and its outflow [50]. An economic and cost-effective way to prevent water pollution and leaching is tillage with crop residues left on the soil surface [48], also called soil conservation tillage.

Further research presents the spatial variability of unsaturated hydraulic conductivity with the mini disk infiltrometer. The variability in the area of 21.7 ha reached up to 4.29 times between the minimum and maximum values [4]. The spatial variability of soil infiltration capacity under additional irrigation with belt irrigators was evaluated on an area of 6.23 ha, where the results showed a significant impact [41]. In addition to hydro limits, soil moisture, and precipitation, their effect on erosive effects must be monitored and addressed when applying additional irrigation and determining the size of individual doses. One of the anti-erosion effects is maintaining a sufficiently high soil infiltration capacity. Extreme high irrigation rates and incorrectly used technology harm the infiltration capacity of the soil. It can cause degradation of the surface part of the soil profile by the kinetic energy of droplets falling on the soil by incorrect water spraying [2].

In situ mini disk infiltrometer measurements were taken in compacted and non-compacted lines. The values of $k$ and their changes in the function of compactness did not show a linear dependence. At some points, the values increased ($x_1$, $x_3$, $x_5$, $x_8$, and $x_{10}$), and in others, they decreased. For this reason, the results obtained from the infiltration measurements with the mini disk infiltrometer were used in a hierarchical cluster (Figure 8), considering the measurements in both lines at all monitoring points. The results indicate a deviation of the three monitoring points from the others that were located in the long-term compression zones. At these points, the deviation between the compacted line and the uncompacted line created by the seed drill just before the measurement is the largest ($1.79 \times 10^{-4}$ cm·s$^{-1}$ in $x_2$, $3.13 \times 10^{-4}$ cm·s$^{-1}$ in $x_4$, and $2.14 \times 10^{-4}$ cm·s$^{-1}$ in $x_6$). In the rest of the results, the dividing line is formed by points $x_9$ and $x_7$.

The map of distances of the mini disk infiltrometer measurement results (Figure 8) present the largest (faintest) and smallest distances between places in colour (dark to zero, the same value black). The results show the variability of hydraulic conductivity and the effect of compaction created by the tracks of the tractor wheels (compressed three bands, Figure 2 on the right) and the effect of the passage of machinery along the line. When using the ANOVA statistical tests with one-factor analysis, all ten monitoring points were included in the study of the influence of the time interval of soil infiltration monitoring; in the case of not only the compacted soil layer by the seed drill ($F_{crit} > F = 2.61$, $p = 0.123$) but also the uncompacted soil ($F_{crit} \geq F = 0.00015$, $p = 0.99$), there was no statistically significant difference. If monitoring points $x_3$ and $x_5$ were not considered with the seed drill compacted soil, the results were already statistically significant ($p = 0.002$, $F > F_{crit}$). In the case of uncompacted soil, a situation arose where the outer monitoring points showed a decrease in infiltration and the central monitoring points increased. From these data, the result of statistical insignificance remains. Again, considering all ten points ($x_1$ to $x_{10}$) when evaluating the statistical dependence on compacted soil by seed drill, the significance

hypothesis was not confirmed ($F_{crit} > F = 2.64$, $p = 0.12$, at level 0.05). The results were not statistically significant when monitoring the dependence of soil infiltration capacity variability from the monitoring point (position on the plot $x_1$ to $x_{10}$) ($p = 0.69$).

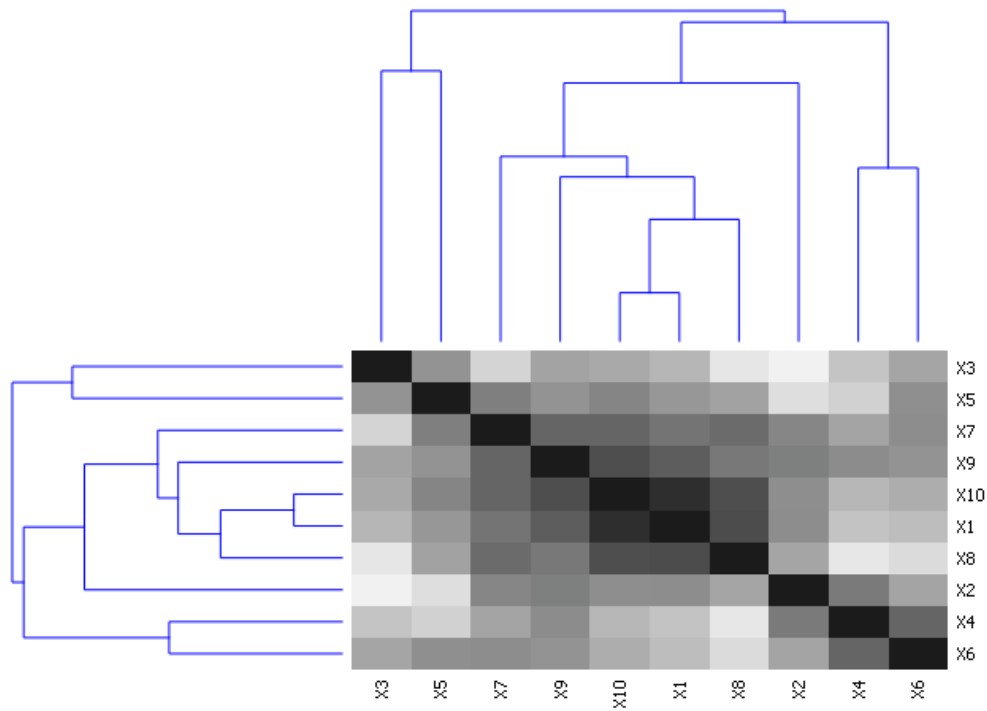

**Figure 8.** Distance map of hierarchical clustering of mini disk infiltration measurement results.

*3.3. Results of Double Ring Infiltrometer Measurements (Method Two)*

When applying the ring infiltrometer, a pair of circles was used as described in the methodology, where measurements were performed at the first three fixed points ($x_1$–$x_3$). As with method one, the MPs were determined at a distance of 0.7 m (compacted soil by selected seed drill, three points) and at a distance of 2.7 m (uncompacted soil, three points) from the auxiliary line. At these six points, the saturated hydraulic conductivity ($k_s$ cm·s$^{-1}$), the standard deviation (cm·s$^{-1}$), and the coefficient of variation $CV$ (%) were determined. The second measurement method did not result in any negative values. Saturated hydraulic conductivity values ranged from $0.15 \times 10^{-3}$ to $13.65 \times 10^{-3}$ cm·s$^{-1}$. The coefficient of variation $CV$ was lower in the case of measurements on compacted soil with one pass by the machinery (58.58% at 900 s measurements). The lowest $k_s$ value was at monitoring point $x_2$ (uncompacted part by the machinery). Monitoring point $x_2$ has also been affected by the long-term cyclical compression of tractor tracks (for more than ten years). Challenging results were figured out where the infiltration rate of the soil compacted by machinery was higher than that of the uncompacted soil (Figure 9).

To evaluate the results obtained by the mini disk infiltrometer and the double ring infiltrometer, it was necessary to convert the unsaturated hydraulic conductivity $k$ to the saturated hydraulic conductivity ($k_s$) using Formula (4) and applying calculation $A$ according to Formula (3). The average value of calculated saturated hydraulic conductivity was $0.2 \times 10^{-3}$ cm·s$^{-1}$ and for uncompacted soil it was $0.41 \pm 0.32 \times 10^{-3}$ cm·s$^{-1}$ (applies to fixed points without the negative values). The results indicate a value of the difference factor (ratio of the value of the saturated hydraulic conductivity of the mini disk infiltrometer to the double ring infiltrometer) in the range from 0 to 0.24 (Table 4). Ghosh and Pekkat [14] reported that the hydraulic conductivity determined by the mini disk infiltrometer was approximately 0.5 to 0.67 times higher than that of the double ring infiltrometer. Radinja [5] reported the average value of the difference ratio factor from the four monitoring points (0.37). Fodor et al. [9] investigated the dependence of measurement methods on hydraulic conductivity. The devices they compared included the equipment we used for our mea-

surements. The results presented that the ratio value (difference factor) depended on the soil type (for sand, 1.1; for silt loam, 0.6).

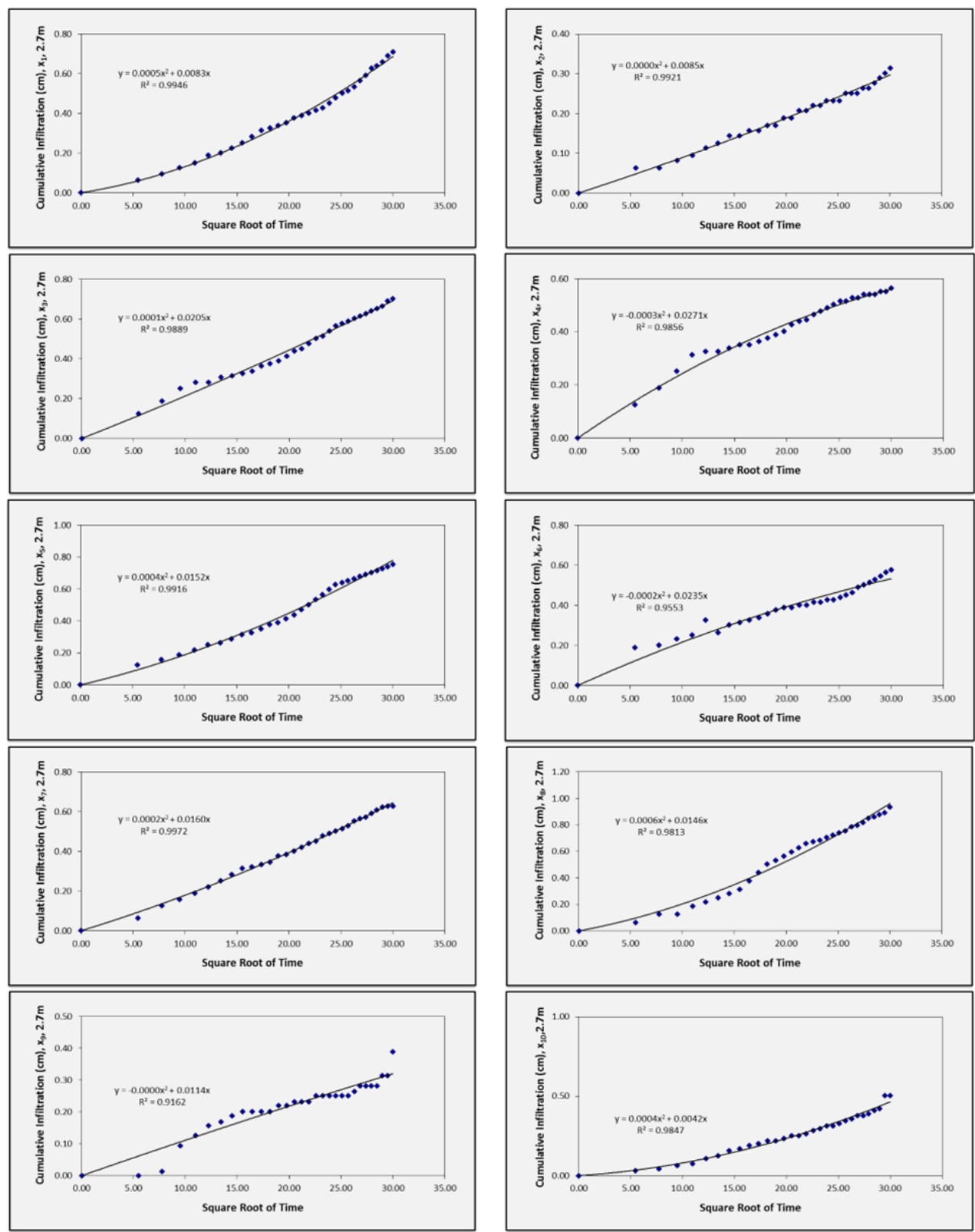

**Figure 9.** Dependence of cumulative infiltration on the square root of time (from $x_1$ to $x_{10}$, 2.7 m, measurement time interval 900 s) determined with method two.

**Table 4.** Hydraulic conductivity measured with the double ring infiltrometer in the monitoring points at 0.7 m and 2.7 m distances.

| MP | $A_1$ | $k_s \times 10^{-3}$ cm·s$^{-1}$ | $A_1$ | $k_s \times 10^{-3}$ cm·s$^{-1}$ | $A_1$ | $k_s \times 10^{-3}$ cm·s$^{-1}$ | $A_1$ | $k_s \times 10^{-3}$ cm·s$^{-1}$ |
|---|---|---|---|---|---|---|---|---|
| | Measurement Data, MB—0.7 m | | | | Measurement Data, MB—2.7 m | | | |
| | T = 300 s | | T = 900 s | | T = 300 s | | T = 900 s | |
| $x_1$ | 0.0028 | 4.20 | 0.0025 | 3.75 | 0.0014 | 2.10 | 0.0049 | 7.35 |
| $x_2$ | 0.0048 | 7.20 | 0.0021 | 3.15 | 0.0022 | 3.30 | 0.0001 | 0.15 |
| $x_3$ | 0.0016 | 2.40 | 0.0003 | 0.45 | 0.02 | 30.00 | 0.0091 | 13.65 |
| Aver. | | 4.60 | | 2.45 | | 11.80 | | 7.05 |
| St. dev. | | 1.98 | | 1.44 | | 12.88 | | 5.52 |
| *CV* (%) | | 43.04 | | 58.58 | | 109.14 | | 78.23 |

Notes: MP monitoring points ($x_1$–$x_3$), *CV*—coefficient of variation.

The results on hydraulic conductivity benefit the scientific team, farmers, and country managers, where the results show how fast water is absorbed in the soil (depending on the kind and type of soil). Transformation is also important in transporting contaminants, groundwater recharge, and ecosystem sustainability. This water movement across the soil usually occurs under saturated and unsaturated conditions, and flowing through unsaturated soil is more complicated than flowing through continuously saturated pores. Soil hydraulic conductivity is strongly dependent on detailed pore geometry, water content, and differences in matrix potential [51,52].

Soil erosion is a major environmental threat to agriculture's sustainability and production capacity. In the last 40 years, almost one-third of the world's arable land has been destroyed by soil erosion [53]. Soil erosion is the most common form of soil degradation. The area of land globally affected by erosion is 1094 million ha, of which 751 million ha are severely affected by water erosion and 549 million ha are affected by wind erosion, of which 296 million ha are in serious condition. Soil erosion is a four-stage process that involves detachment, collapse, transport (redistribution), and sediment deposition. Thus, soil erosion strongly impacts the global carbon cycle (carbon dynamics) [54,55].

As it can be seen from the overall review of the literature and the results, infiltration is one of the key processes in modelling runoff and designing measures to regulate rainwater in urban areas and agricultural production. Estimation of hydraulic conductivity and fast methods of obtaining results are helpful for the deployment of other work activities in agricultural production.

## 4. Conclusions

The study was aimed at evaluating the hydraulic conductivity of the soil. It was observed that both methods point to the variability of the results within the position of the monitoring points and thus also within the compacted and non-compacted parts of the land. The difference compared to other results from other authors is that the research was carried out on agricultural land, where we chose three forms of soil compaction (uncompacted, compacted only in the lines by the passage of the machinery, compacted not only in the lines but also transversely by the passage of the tractor—wheel next to wheel method). In some places, the phenomenon of water repellency appeared, which could be caused by the drier location of the targeted plot. However, we managed to alleviate the mentioned phenomenon by extending the measurement time. The largest differences in hydraulic conductivity between compacted and non-compacted soil were demonstrated at sampling points in transversely compacted lines. The obtained results show that the degree of soil compaction reduces the ability of the subsequent infiltration of water into the soil. The following conclusions can be drawn from the achieved results and established hypotheses:

- The used infiltration monitoring method showed differences in the measured values, with the difference factor reaching values of up to 0.24. The results obtained

by the double ring infiltrometer method indicate higher values than the mini disk infiltrometer measurements.

- From the point of view of monitoring the statistical significance of the location of the soil infiltration capacity measurement, the hypothesis was not confirmed. However, when evaluating the results with the first method (mini disk infiltrometer) in a hierarchical grouping in both lines and at all ten monitoring points, the results show a substantial deviation of three monitoring points from the others, which were located in the zones of long-term soil compaction.

- The time interval of the soil infiltration capacity monitoring showed differences in the case of compacted and non-compacted soil; however, they were not significant when statistically considered. The hypothesis regarding the phenomenon of water repellency has been confirmed, which means that the effect of water repellency can be reduced by extending the measurement time interval.

**Author Contributions:** Conceptualization, J.J. and K.K.; methodology, J.J. and M.A.; software, J.J., K.K. and M.A.; validation, J.J., K.K. and M.A.; formal analysis, M.A. and K.K.; investigation, J.J. and M.A.; resources, J.J.; data curation, J.J. and K.K.; writing—original draft preparation, J.J.; writing—review and editing, J.J., K.K. and M.A.; visualization, J.J.; supervision, J.Z.; project administration, J.J. and J.Z.; funding acquisition, J.J. All authors have read and agreed to the published version of the manuscript.

**Funding:** This publication is the result of the implementation of projects: (1)"Scientific support of climate change adaptation in agriculture and mitigation of soil degradation" (ITMS2014 + 313011W580), supported by the Integrated Infrastructure Operational Programme funded by the European Regional Development Fund (ERDF), and (2) APVV-20-0071 funded by the "Slovak University of Agriculture", Nitra, Tr. A. Hlinku 2,949 01 Nitra, Slovak Republic under the projects 'APVV-20-0071'.

**Data Availability Statement:** This study did not report any data.

**Acknowledgments:** The authors are grateful to the staff of the University Farm in Kolíňany, Slovakia, for their technical and operational support during this research.

**Conflicts of Interest:** The authors declare no conflict of interest. The funders had no role in the design of the study; in the collection, analyses, or interpretation of data; in the writing of the manuscript; or in the decision to publish the results.

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
