# Peer review of "Evaluation of Soil Infiltration Variability in Compacted and Uncompacted Soil Using Two Devices"

_water, doi:10.3390/w15101918_

Round 1
Reviewer 1 Report
1- The Abstract has written very good and it does not need any changes. The keywords that authors have used in the manuscript should be different from those keywords which have been used in TITLE.
2- The first letter of all keywords which have been written in the manuscript should be capital.
3- In introduction, which has written very good, the paragraphing system is not appropriate. The beginning of each paragraph should be started with the new topic and idea.
4- In Materials and Methods, for lines 130-134, and for references 19 and 20, and for references 21 and 22. It is recommended to clearly illustrate and it is also better to design one or two tables related to this information (Crop variability, soil strength, soil properties, and the temperature parameters) for it.
5- For figure 2 and other figures and tables, according to the MDPI format, just Figure and Table should be BOLD, not all other sentences. Figure three is OK.
6- Lines from 313 to 319 is not correct. It is better to delete unnecessary information from RESULTS to Results of Mini Disk Infiltrometer measurements (method one).
7- The part 3 which is RESULTS has written very well, and I think Discussion has been included in the part RESULTS, although, the title is NOT Results and Discussion. It is strongly recommended to separate those information in Result section which is related to DISCUSSION in one separate part, headline 8. Please, check all references format. For example, References number 16 and 21 and 17 and 28 and 47 and 48 and 52 are not CORRECT at all. All references should be checked with numbering system in the manuscript.
(Please, pay very careful attention to referencing systems and use DOI and write them at the end of all references)
Author Response
Please see enclosed document where all the comments and recommendations were addressed.

Reviewer 2 Report
The paper is interesting but poorly supported with data. In Methods the approach is so simple. The analysis not include the spatial analysis with results simple. In general, the paper is simple, with no interes to readers and it can be substantially improved (more data or change in methods).
Author Response
Please, see the enclosed document where all the comments and recommendation were adressed.

Reviewer 3 Report
In this paper, author studied “Evaluation of soil infiltration variability in compacted and un- compacted soil using two devices”
This is an interesting manuscript, and it brings some interesting and useful information, and I think that the manuscript has a potential to be published in the journal. However, there are some recommended amendments are required as follow:
Line 25-26: the sentence started with “A one-factor … should be removed from the Abstract section to M&M section.
Line 22 – 25: The results presented in the abstract section is not representative, please add more detailed results as for what is presented at the conclusion section.
Line 91-99: By the end of the introduction section, here author should present a clear “detailed” objective of the study and also to have a clear hypothesis.
Line 314-319: Please remove this part
Line 321-326: these information is not part of your results, though it should be moved to M&M section.
The results section should be amended to present the results from the used methods (method 1 and two) simultaneously in order to have a clear comparison between both methods. Additionally, the discussion part is not enough for the presented results.
The conclusion section is too long including unnecessary parts describing the methods used and/ or detailed results. This section needs to be rewritten.
Author Response
Please, see the enclosed document where all the comments and recommendation were addressed.

Round 2
Reviewer 1 Report
The article can be accepted in present format.
Author Response
Dear reviewer,
thank you for your efforts regarding our manuscript. We did our best to improve it according to your comments.
Thank you and best regards.
Reviewer 3 Report
Thanks for your effort performing the recommended amendments. The Manuscript is now much more in better form for publication
Author Response

(The authors gave the same response as above.)
